# Collections in the Expanded Field: Relationality and the Provenance of Artefacts and Archives

**Michael Jones** 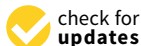

eScholarship Research Centre, The University of Melbourne, Parkville, VIC 3010, Australia;
m.jones@unimelb.edu.au; Tel.: +61-431-331-537

**Abstract:** In 2017 archaeological evidence was published which indicates that modern humans first arrived in Australia around 65,000 years ago. Through the countless generations since, Aboriginal and Torres Strait Islander peoples built deep connections to the landscape, developed rich material culture infused with story and myth, and used oral and ceremonial traditions to transmit knowledge over thousands of years. Yet, since European invasion at the end of the eighteenth century, the provenance of ethnographic and institutional collections has largely been documented with reference to white collectors and colonial institutions. Attitudes are starting to change. Recent decades have seen significant moves away from the idea of the authoritative institution toward relational museums and the co-creation of knowledge. But the structure and content of much museum documentation continues to lag behind contemporary attitudes. This paper looks at the documentation of Australian ethnographic and anthropological collections through the lens of changing perspectives on provenance, including archival notions of parallel and societal provenance. When placed in the context of recent developments in material culture theory, these collections help to highlight the limitations of existing documentation. The paper concludes with a call for community involvement and a more relational approach to documentation which better encompasses the complexities of provenance and the entangled institutional, archival, oral, and community perspectives that accumulate around artefacts in museums.

**Keywords:** museums; provenance; collections; documentation; archives; Australian history; Aboriginal and Torres Strait Islander history; anthropology; ethnography

## 1. Introduction

In 1978, the British Museum found an unregistered Aboriginal shield in its collection and gave it a number: Oc1978, Q.839 [1]. Made of bark and wood, the narrow shield, tapered at the top and bottom, stands nearly a metre tall, the front rough and blackened, the back smoother with a thin handle attached. A hole near the centre suggests damage from a spear, or perhaps a bullet. There is a story here; but no documents have been found linking the artefact to a specific collector, or to other items in the collection, nor are there any extant records in the museum regarding its acquisition or provenance. There is, however, a small nineteenth-century label on the back which has been identified as written by traveller, collector, anthropologist, and museum volunteer James Edge-Partington. It reads: "CAP.COOK." [2]. In addition to navigator Captain James Cook, associated names in the online collection record include Sir Joseph Banks, and their ship the HMS *Endeavour*. There is only a passing mention of the local Gweagal People who likely made and used the shield, in the Curator's Comments sandwiched between a short bibliography and the results of testing carried out by British Museum scientists.

Museums frequently privilege collectors, scientists, and 'great figures from history' when documenting artefacts, particularly those taken from the colonised and dispossessed. For ethnographic

artefacts in particular, documented provenance often starts with collection by Australian, European, or American anthropologists and expeditions rather than with the names or stories of those who designed, manufactured, or used the item. This paper explores the broad concept of provenance in the context of museum collections and archival records, drawing on archival theory, museological and curatorial investigations into the 'relational museum,' and case studes from the history of collecting in Australia. 'Provenance' is here viewed as one element in an expanded relational field [3] of description which can (and should) contain other possibilities for the theory and practice of museum documentation. In doing so, museums can better reflect complex, entangled histories, and the multiple contexts and knowledge systems that surround artefacts and other collection items.

As a white academic I do not write on behalf of Indigenous Australia. With hundreds of communities speaking more than 250 langauges, the Australian continent contains a huge diversity of cultures and perspectives, including many different attitudes towards museums, archives, and other cultural heritage institutions. Here I argue that those of us embedded in European institutions must not assume we have the experience, or the right, to make decisions regarding provenance, knowledge, and the representation of First Nations artefacts and ideas. Instead, we must critique existing practice as the first step toward engaging with communities directly, seeking to develop more complex, polyvocal aproaches to provenance in museum documentation (and in archives, libraries, and galleries), stepping aside to let others speak, and relinquishing the perceived certainties of institutional authority and the pursuit of singular truths.

The first section provides an overview of provenance in museums, archives, and more broadly, highlighting developments which start to complicate past practice. Following this is an outline of the development of the 'relational museum,' which has contributed to a growing conceptual basis for more complex, multi-layered documentation. Section 4 looks at a number of specific examples, including the Gweagal shield, to highlight the complexities of provenance and the privileging of dominant cultural perspectives over Indigenous and First Nations knowledge. A discussion section then uses the previous sections to further explore the limitations of museum documentation practice, and to suggest strategies for change.

## 2. Provenance in Museums, Archives and Beyond

The way provenance is defined and described varies within and across institutions and disciplines. While a detailed exploration of these differences is beyond the scope of this paper, some elements are worth highlighting to provide a foundation for the sections that follow.

In museums, provenance is often used to refer to the source or origin of an item. This can include the manufacture or creation of artefacts and objects, acquisition or donor information, and (particularly for artworks) may include the history of the ownership, custodianship, and display of the item. A policy statement from Museums Victoria (Melbourne, Australia) is in keeping with this tradition, defining provenance as: "the full history and chain of ownership of an item from the time of its creation or discovery through to the present day, from which authenticity and legal ownership is determined" [4].

But concepts such as 'discovery,' 'ownership,' 'authenticity,' and 'legality' are complex and potentially contested. For example, the British Museum's shield is now the focus of repatriation claims by descendants of the Gweagal warrior Cooman, who is thought to have been its bearer when Cook landed in 1770 [5]. The International Council of Museum's CIDOC (Comité International pour la Documentation) Conceptual Reference Model wisely treats legal ownership (E8 Acquisition) and physical custody (E10 Transfer of Custody) as separate [6], but concepts of legal ownership are deeply problematic for longstanding collections, particularly those acquired during the invasion and colonisation of nations with quite different systems of ownership and law. As archivist Michel Duchein writes, principles like provenance are often "easier to state than define and easier to define than to put into practice" [7].

Archaeological artefacts can have both a provenance and a provenience. The latter refers to the record of an object as it was found in situ (its 'findspot'), including its spatial coordinates, relationship to other objects, and its condition [8]. Together, provenance and provenience theoretically allow people to trace an item back from its display case to the exact physical context of its unearthing, and from there into stories of its creation and use. In practice, long-standing collections may suffer from lacking or lost documentation in this regard. Archaeologist Payson D. Sheets writes: "The artifact with no context or provenience is virtually worthless [ . . . ] Tragically, their very presence in collections and museums indicates the impossibility of ever gaining solid contextual information about them" [9].

Thinking about digital data provides an interesting comparison. The 'findspot' in a collections management database or similar is a field containing particular data, with "where-provenance" recording the source of the data, and "why-provenance" recording the reasons for their presence in the database [10]. Although not explicitly acknowledged in the literature, provenience would be an equally useful concept for databases, recording the location of an element within a broader data structure, its relationship(s) to other elements in the database, and so on.

Archival provenance is related to museum and archaeological provenance but is also in some regards a field in its own right, with a long history of debate as to the nature of the concept and its actual or intended implications for practice. Canadian archivist Shelley Sweeney writes: "in the archival field provenance is the basis not only for establishing authenticity and sometimes for dating records but also for their acquisition, appraisal, arrangement and description, and retrieval. Thus, what constitutes provenance is critically important to the archivist" [11]. From the earliest days of the profession the creator and source of records—whether an organisation or government, or an individual—has been the primary determinant of the way archives are kept and documented [12,13]. Aggregates of records are (at least theoretically) maintained in the order in which they were created and used, and records with one provenance are not intermingled with records from another source.

Over time, discussion of provenance has increasingly focused on relationships. Archivist and humanities technologist Daniel Pitti and colleagues [14] have written of how, historically, the "network of interconnections between people" responsible for creating and maintaining archives and records was embedded in archival description. Archivists have worked toward the structural separation of these elements. Australian archival science has been particularly strong in this regard, starting in the 1960s with a move toward capturing provenance as distinct entities which are then related to aggregates of records (see, for example, [15,16]). Chris Hurley, Tom Nesmith, and Michael Piggott have all built on these ideas to explore ideas about parallel and societal provenance, broadening and enriching notions of provenance to include multiple perspectives and the shared involvement of different individual and collective provenance entities in the creation and co-creation of records [17–20].

Laura Millar is one who has has argued for combining the areas outlined above: chronological museum and art historical provenance; archaeological provenience; and archival provenance. She outlines three interconnected areas of focus: (1) creator history, or "the story of who created, accumulated, and used the records over time," focused on the creator and not the records; (2) records history, or "the story of the physical arrangement and movement of the records over time. This would be an adaptation of artistic provenance"; and (3) custodial history, or "the explanation of the transfer of ownership or custody of the records from the creator or custodian to the archival institution and the subsequent care of those records" [8]. Capturing these ideas together requires a relational approach, connecting various elements to build a complex representation of the context in which records (and, by extension, any things) emerge, move, and develop through time. As suggested by Terry Cook, archival aggregates are created through relationship descriptions, with provenance lying "at the heart" (Terry Cook in [21]).

### 3. The Relational Museum

Over the same period, museums were also increasingly interested in relationality. The term 'relational museum' has been used to refer to a composite of trends and perspectives, as outlined by museologist and museum planner Duncan Grewcock:

> For some time now, the academy and the museums profession have been coming to terms with new ways of thinking and representing a more complex, partial, processual world of connections, a world that does not sit so easily within these modernist regimes of classification (if it truly ever did). Recognising and working with a partial and shifting understanding of the world informs the emergence of what one can term 'the relational museum.' The relational museum emerges through varying attempts to re-image the contemporary museum as connected, plural, distributed, multi-vocal, affective, material, embodied, experiential, political, performative and participatory. [22]

This represents a shift away from the relationships of classification, taxonomy and disciplinary knowledge, those regulatory processes that had suppressed the heterogeneity and complexity of museums and collections [23,24] from the birth of the modern museum to the mid-twentieth century.

Following early work to rethink the role of museums in the 1970s (see, for example, [25,26]), the 1980s saw the emergence of 'new museology.' This embraced recent shifts in the politics of representation, and an increased awareness that knowledge and values are contextual and contingent rather than universal. Broadly speaking, new museology focused less on the administrative processes and professional practices found in museums, and more on the place of museums in society, the contextual and historical nature of museum knowledge, and the multiplicity of social, cultural, and political environments which shape our understanding of objects [27,28]. If objects themselves are taken as mute, meaning is historical and contextual (that is, relational) rather than inherent or fixed [29–33].

As contexts change, the meaning and significance of artefacts also develop and change. Viewed diachronically, this perspective leads to the idea that things have biographies which can be explored, and which continue to evolve beyond the point when an artefact has been collected and stored by an institution [34,35]. Viewed synchronically, artefacts have multiple meanings simultaneously. In the introduction to *Exhibiting Cultures*—the first of three influential anthologies co-edited by Ivan Karp—Steven Lavine and Karp note the need for exhibition design which allows for the presentation of multiple perspectives, or "to reveal the tendentiousness of the approach taken" [36]. Sometimes these meanings will compete with, resist, or even contradict each other. Understanding the plurality of meaning requires admitting diverse voices. Much of the movement in this space centres on anthropological and ethnographic collections [an issue discussed throughout the anthology *Museums and source communities*, [37], see also [38,39], and the need to move beyond what anthropologist and museologist Christina Kreps calls "the hegemony of the management regimes of Eurocentric museology" [40].

### 4. Australian Artefacts and Their Provenance

Australian ethnographic artefacts provide ideal case studies for exploring the complexities of provenance and the relational museum. Much of the archival thinking referenced above has emerged from Australia (Scott, Cunningham, Hurley, Piggott) and Canada (Sweeney, Nesmith, Millar, Cook), where colonisation and the rights of First Nations communities complicate received notions of linear, authoritative, or singular statements about provenance; and much of the literature of the relational museum is related to ethnographic and anthropological collections. But while exhibition spaces and scholarly writing about anthropology, archaeology and related disciplines include ideas of entanglement, networks, and meshworks (see, for example, [41–46]), and collection documentation often continues to reflect the limitations of past practice.

Returning to the Gweagal shield, the evidence for Edge-Partington's claim that it is associated with English navigator Captain James Cook is not known. However, piecing together illustrations and contemporary accounts, many believe the shield to be one of those used by two Gweagal men who opposed Cook and his crew when they landed at Botany Bay on 29 April 1770. The shield was presented in this historical context in the 89th episode of *A History of the World in 100 Objects*, and in both the *Indigenous Australia: enduring civilisation* exhibition at the British Museum in London (2015) and the subsequent *Encounters* exhibition at the National Museum of Australia in Canberra (2015-2016) [47,48]. While some, like museum director and curator Nicholas Thomas, question the shield's provenance [2], it has become a cultural touchstone, symbolising the violent dispossesion of Aboriginal Australians by Europeans and highlighting the inextricable link between colonisation and collecting. In the words of Indigenous artist Jonathan Jones: "We do know that Australia's collection methodology started with Captain Cook stealing shields after shooting at someone. I've always used that as a bit of a benchmark for the acquisition process of this country" (Johnathan Jones in [49]).

The collection description provided online by the British Museum acknowledges the shield's uncertain provenance. It is listed as "Found/Acquired: Botany Bay (?)," and the curator's comments begin: "Possibly obtained on Captain Cook's first voyage (HMS Endeavour 1768-1771) on April 29 1770 at Botany Bay, in present day New South Wales, Australia." All these place names are European. Beneath a short bibliography the Curator's Comments do note "that the southern shore of Botany Bay was known as Gwea, and therefore the people from that area called themselves the Gweagal," but even this comes from "Accounts from the early period of European settlement." Meanwhile, the ethnic name in the fielded metadata remains unhelpfully broad given the hundreds of nations which existed on European arrival: "Made by Aboriginal Australian."

The following names are included as fielded metadata.

Associated with: Sir Joseph Banks (?)
Associated with: Captain James Cook (?)
Associated with HMS Endeavour (?)

Beneath this, the acquisition name reads: "From: Sir Joseph Banks (?)." The specifics of the shield's provenance remain focused entirely on the names of European people and a famed ship. This despite the fact that the Gweagal People have passed down stories of these early encounters with Europeans through oral tradition to the descendants of the warrior Cooman, who they believe to be the owner of the shield [5]. Gweagal tradition tells of European arrivals firing on the locals, Cooman dropping his shield as a result; the collection record favours test results from British Museum scientists and a quote from Joseph Bank's journal which suggests the shield "had been pierced through with a single pointed lance." The 'truth' of these seemingly-contradictory statements regarding events from almost 230 years prior is all but irrelevant. At present the museum's public collection documentation privileges European people, place names, and perspectives, rendering alternative narratives (including those of the shield's maker and owner) invisible. The artefact and its significance have been colonised.

In other contexts the elision of Indigenous provenance can be even more complete. A shield on sale at Sotheby's on 21 September 2016 was given as: "A Broad Shield, Lower Murray River, South Australia 19th Century. Carved and engraved wood (inner bark of the gumtree), bentwood handle, 88 cm by 26 cm. Provenance: Private collection, France" [49]. Wiradjuri man and Indigenous Project Officer at the Australian Museum, Nathan Sentance, writes of this tendency to privilege white collectors and donors in collections documentation:

> All of these examples decentre First Nations people. They also imply that First Nations knowledge or culture doesn't exist until it gets white acknowledgement. That our culture, like our land, needs to be 'discovered.' Furthermore, it doesn't recognise First Nations people

as creators of culture and history or as knowledge holders, but rather gives them the roles of subjects. [50]

Aboriginal Australians represent at least 65,000 years of continuous culture [51], yet collections in galleries, libraries, archives, and museums (GLAM) remain "filled with White voices" [50].

There are countless other examples, including from groundbreaking expeditions which remain significant to this day. An early example is the Cambridge expedition to the Torres Strait Islands in 1898, led by Alfred Cort Haddon. Coming at the end of decades of 'armchair anthropology' [52] (pp. 15–16) where field collecting was left to travellers and locals working on behalf of scholars [53], the innovative team spent seven months in the Torres Strait reconstructing ceremonies, checking and cross-checking information with informants, and capturing audio and film recordings [54,55]. Another aspect considered progressive was Haddon's acknowledgement of local sources. Anita Herle writes:

> In contrast to the practices of many early ethnographers, Haddon took great care to acknowledge the sources of the Expedition's data. Islanders and other informants were referenced and often quoted verbatim, many of the people in the photographs were identified, and the names of an object's maker or previous owner were sometimes recorded. [56]

Among the many artefacts collected by Haddon, and now held in Cambridge's Museum of Archaeology and Anthropology, is a large mask made from the whole shell of a hawksbill turtle, with a wooden nose, and painted eyes, cheeks and mouth. Shells are attached at the top of the forehead and under the chin, with the whole fringed by feathers and vegetable fibres.

However, although part of the Haddon collection, the turtle shell mask is not one of the 1800 items Haddon collected on his two expeditions. The details in David R. Moore's descriptive catalogue [54] include two limited statements regarding provenance, and some references:

365. Z.10040. Plate 42
Mask, human face, turtle-shell, *dogaira wetpur krar* or *le op*.

> Large mask made of whole shell of hawsbill turtle, fringed with cassowary and other feathers and vegetable fibres. Attached nose of wood. Nose, eyes, cheeks and mouth painted red and white. Cowries attached at forehead and chin. Crossed bullroarers as crest. Used in dances celebrating a good harvest. Stated to have been made to order, smaller than original.

L. 76 cm; W. 50 cm.
J. Bruce, Mer (1905?).
*Refs*: TSR 1: 179-81
TSR 6: 209. pl. XXII, [ . . . ]
Haddon Archive (CUL) 1036.

The references are to specific pages within 'TSR' (referring to the *Reports of the Cambridge Anthropological Expedition to Torres Straits*) volumes 1 and 6; and envelope 1036 in the Haddon Archive, held by the Cambridge University Library (CUL), which contains a field book once belonging to J.S. Bruce.

Historians of collections or early anthropology are given only a few tantalising clues. If this particular mask is "smaller than the original," where is this original and how big is it? Moore's catalogue says it is "stated to have been made to order." Stated where, and by whom? Are there other items made specifically for Haddon and his associates, or is one of only a few? Moore notes in his introduction to the collections:

> Obviously by the time he was in the Strait many native materials had gone out of use, but even so Haddon encouraged the Islanders to manufacture artefacts 'in the old-time fashion' especially for him, and often the older men and women were delighted to do this. [54]

This seems to suggest many items were, in a sense, 'made to order,' but Moore does not refer to other parts of the collection in the same way. As for the online catalogue, it now reads "Made for J. Bruce," which is only slightly more useful. WIthout effective data provenance and provenience, the source of this claim and its relationship to other sources remains unclear. Searching through the *Reports*, the online catalogue, and other related material reveals that Bruce was a European informant on Mer (also known as Murray Island) who worked with Haddon over many years, but with no relationships in place every user must (re)discover this for themselves. Perhaps the end of the trail is Bruce's field book, archival item 1036, which notes that an earlier case of items sent to Haddon in December 1903 never arrived. Its contents had included "1 turtle shell mask of Dogai," as did the replacement case sent in 1905. It seems the smaller mask was ordered to replace one lost in transit. Information on who made the mask, or who helped to interpret its significance and purpose, is entirely absent, as it is for other items which ended up in Cambridge. The provenance of the material is generally given only as "Haddon, Dr. Alfred Cort [collector]" or variations thereon.

Australian anthropologist Donald Thomson is another recognised for pushing the boundaries of the discipline. Howard Morphy has suggested that Thomson's work from the late 1920s to the 1960s was something of an anachronism:

> It maybe that Donald Thomson was an anthropologist out of his time, because when Donald Thomson was working, anthropology was going through a period where it was much more interested in social organization, in kinship, than it was in material culture and art. Thomson was passionate about material culture, passionate about art objects. But no one was really interested in them and interested in his writings about them.

(Howard Morphy in [57])

Thomson also felt an outsider in white Australian society more broadly; he describes the moment when, in 1937, HMAS *Moresby* came to pick up him and his collection from the Northern Territory: "I felt old, very tired and very dirty. I realized I did not want to go back to civilisation, that I knew and loved the Arnhem Land people and that I had more in common with them than my own kind" (Donald Thomson in [57]).

His perceived affinity for Aboriginal people meant Thomson gained a reputation as progressive. Building on Haddon's practice of naming his sources, Nicolas Peterson considers Thomson's writing "unique in that it names Aboriginal people, presenting them as individuals and active agents in local history in a way few other writings do" [58]. Yolungu elder Wirilma Munungurr recalls: "The people of Arnhem Land said they liked Dr. Thomson, because he thought like an Aborigine. They were very close and listened to each other" (Wirilma Munungurr in [57]).

In other ways, Thomson followed anthropological tradition. He collected human remains, and (in a practice that mirrors Haddon's pursuit of pre-contact artefacts) he requested that people remove their clothes and any other signs of European influence before they were photographed [59], as though this would capture a more 'authentic' representation. Alec Naiga recalls this occurred when Thomson came to photograph the dugong hunters of eastern Cape York. The published photographs from this visit also leave out any sign of the busy port that operated at the mouth of the Stewart River, the surrounding pastoral land, or other features which might reveal the impact or extent of European settlement [59]. There have been suggestions his photographic practices were as much about not wanting to show "his Aboriginal friends as 'degraded' outcasts," [59] and historian Paul Turnbull provides evidence for elders from relevant clans stating that the ancestral remains acquired by Thomson would have been given to him "for good reasons" by community members who held him "in high regard" [60]. Whatever the motivations, the result was the same: photographic representations of Indigenous peoples shaped by an ethnographic eye, and more human remains entering museums, with all the potential problems around discovery, access, and repatriation that entails.

These are complex stories of creation and acquisition. Local Indigenous communities were involved, not just as subjects of study, but as active participants. Some people demonstrated

the manufacture of artefacts, or took Thomson on hunts, imparting detailed knowledge about cultural practices and the local flora and fauna. Others made items on request, or because they did not want to part with the things they used in everyday life. These encounters, documented by Thomson in thousands of pages of notes and beautifully-composed (if sometimes staged) photographs, shaped the knowledge and items that ended up in museums. While curators, scholars, and communities continue to work with and through these entangled encounters and the traces which remain, publicly accessible collection documentation often remains sparse by comparison. References to communities and language groups are included, but the Collection Name remains "Donald Thomson Collection," the Acquisition information reads "Long-term Loan from University of Melbourne (The), 28 March 1973," and the Maker is frequently listed as "Unknown."

Art, artefacts, landscape, and intangible heritage are equally open to being claimed. Anthropologist and geographer Marcia Langton addressed the International Council of Museums conference in 1998:

> The supremacist vision of Australian nationhood dominates Australian public culture. This [ . . . ] is apparent in the appropriation of all things Aboriginal, sacred and profane, as a form of cultural wallpaper decorating the back stage. Sacred dreaming designs have become the Olympic Games logo; the magnum opus of the late Kngwarreye, the painter from the central desert, has become a travelling exhibition; and the sacred didjeridu sound is on inflight entertainment. Claims to national authenticity by very recent Australians are everywhere underscored by emblems of our genocide.

> (Marcia Langton in [61])

Even natural history privileges colonial agents. Around the time Haddon was in the Torres Strait, Baldwin Spencer and Frank Gillen were engaged in their own extended fieldwork in Northern Australia. Two species of monitor lizard collected by Aboriginal field informants were named *Varanus spenceri* and *Varanus gilleni* in their honour [61] (p. 78). The informants' names are lost to history, and the natural world tagged as a European 'discovery.'

## 5. Discussion

Exploring the ethnographic and anthropological collections of any large museum will reveal examples comparable to those above. Museums, far from being neutral observers and documenters, can contribute to the nationalist vision Langton describes, passively accept the historical privileging of white colonial perspectives, or seek to become agents for change. Some have done more than others to document provenance, but there is invariably still more work to do. More effective documentation of provenance and related contextual information requires a number of interconnected elements.

For museum artefacts, 'acquisition' is an event involving multiple parties, sometimes spanning long periods of time, not a one-to-one relationship between institution and donor. As with the archival tradition, noted by Daniel Pitti et al., descriptions of these events, entities, and processes need to be disembedded from item-level descriptions to create networks that better reflect the relationships and flows between people, communities, and organisations. Where these stories relate to multiple artefacts or large collections, it makes sense to document provenance at aggregate level and only describe the provenance of individual artefacts where they differ from or add to the broader context.

Provenance should also be more than just the history of the item. Expanding on Millar's call for creator histories, records histories, and custodial histories, details could include: the histories of people, organisations and events; the histories of things and their relationships; and the history of custodianship and exchange. Archival notions of societal provenance are useful here in broadening the scope of what is considered, going beyond canonical individuals (Cook, Banks) and colonial voices to incorporate community perspectives and knowledge. As Tom Nesmith notes: "Societal provenance is not just another layer of provenance information to add to other ones such as the title of the creator(s), functions, and organizational links and structures. The societal dimension infuses all the others" [19].

Incorporating alternative voices and perspectives involves an explicit recognition that First Nations people are creators and bearers of culture, history, and knowledge, not merely the subjects of study. As an elder told Sentance: "Museums have the sticks, we have the stories. Without the stories, museums only have sticks" [62]. With this shift comes a recognition of the limitations of our own perspectives. Recent Australian exhibitions have taken the step of referring to the creators of items as "Once known," or as "Made by Ancestors" [63,64], highlighting that lack of knowledge is itself temporally and contextually situated, not universal. Capturing these different perspectives in documentation reflects the polyvocal concepts increasingly encountered in other parts of the relational museum, and beyond.

Once provenance and context are conceived relationally, the possibilities continue to expand. Jeanette Bastian's notion of the "provenance of place" [65] is useful when considering the connection between artefacts landscape, and Indigenous country in Australia (and elsewhere). Kungarakany and Arrernte man Shaun Angeles Penangke spoke of this in his keynote at the 2018 National Digital Forum (Wellington, New Zealand), describing how the creation of Arrernte collection management systems included linking everything back to a place or a site, capturing their way of looking at the world [66].

Data provenance provides a framework for critiquing the lack of citation and explicit evidence attached to attributed data in collections management systems. In keeping with Lavine and Karp's call to acknowledge the tendentiousness of particular approaches, relationships allow multiple perspectives on discovery, ownership, authenticity, and legality to co-exist without needing to resolve all disputes into a single, supposedly-factual statement. For some this may seem a step too far. Author and attorney Rafia Zakaria has argued that museums "fight against having to reveal the provenance of objects precisely because they know that many of the objects in their vast collections have illicit histories behind them" [67]. But relationships to evidence for each claim has a role to play here too. Much of the evidence required to determine or demonstrate complex provenance relationships are contained in archives and records, which are themselves richly-relational aggregates with their own provenance and context. As Penangke notes, all of the assets in a collection are "interconnected, and you can't have them sitting in silos. You can't isolate them from one another because it just doesn't make sense, the collection doesn't make sense" [66].

Providing explicit relationships to archives, images, and other records is a significant part of making knowledge accessible back to communities. Lyndon Ormond-Parker and Robyn Sloggett write of the value of this material:

> Institutional records may hold information that no longer exists in the same way in community. One example is museum archives relating to anthropological and scientific expeditions; the photographic records sit with informant records and objects as part of the 'evidence' collected during this expedition. This historical material can be very important in reconstructing and passing on knowledge at community level. [68] (p. 194)

But other sources are equally significant. Oral tradition is used to preserve and pass on knowledge by First Nations communities in Australia, where stories have been shown to reference events from up to 10,000 years ago [20,69]. As with the Gweagal shield, this includes perspectives on encounters and confrontations no less significant than the journals of people like Banks. Not all such knowledge works counter to the custodial focus of many museums. Effective links to a variety of evidence can support claims to institutional custody (where those claims are well-founded) as much as challenge them.

There are inevitably challenges associated with the idea of more complex, relational, and polyvocal collections documentation. As Morphy notes: "one of the difficulties in studying material culture is that once one begins to trace the network of connections that are linked to a single object it is difficult to know where to stop" [70]. Resource limitations may have a slowing effect here, though without effective processes in place the descriptions for heavily-used items become more

tangled than entangled. Over time, and based on findings from practice, those responsible for museum documentation and provenance will need to work toward the elusive goal described by archivist Victoria Lemieux, who hopes for "parsimonious, yet sufficiently expressive, approaches to representing archival contextual complexity" [71].

The fact that complex, relational museum documentation is difficult should not in itself be a basis for avoiding change. Making what we know about items more discoverable—Including contested interpretations, problematic collection practices, networks of distributed archival and published materials, and the research material (whether voluminous or worryingly spartan) found in curatorial item files—is as much part of serving our diverse publics as the provision of some clear images and a neat set of fielded metadata. Where artefacts and related records come from colonised communities, the need to open new doors and stand aside becomes even more pressing [72]. This is part of the broader decolonisation of institutions and practice [73]. Digital technologies have a significant role to play. Ormond-Parker and Sloggett have looked at some of the ways in which technology supports Aboriginal empowerment with regard to cultural heritage [68], and museum documentation systems have the potential to contribute to this broader shift.

In addition to more complex, relational approaches to provenance and collection documentation, fully supporting this change requires the involvement of Indigenous communities at a foundational level.

First, databases and standards need to be reformed. As Michael Christie notes, "Databases are not innocent objects. They carry within them particular culturally and historically contingent assumptions about the nature of the world, and the nature of knowledge," many of which reflect long-standing traditions of Western thought. He continues:

> An indigenous database must be a lot more than simply a conventional database full of representations of Aboriginal knowledge. For it to be an indigenous database, its architecture and structure, its search processes and interfaces, its ownership and uses must also reflect and support indigenous ways of being and knowing, and their control over their own knowledge. [74]

Christie argues that, in contrast to 'facts' assigned to fixed fields in many existing databases, Aboriginal knowledge is based on "producing, prosecuting and assessing situated and timely truth claims" [75]. Capturing this type of knowledge is fully in keeping with a more relational, polyvocal approach to documenting artefacts and other entities (people, places, communities, events, stories, documents, specimens, observations, and more), which in turn leads to an ability to capture alternative knowledge systems [76], fluid ontologies [77], and moving narrative lines [46,78] over the fixed classification of discrete things.

Second, redeveloping our systems, processes, and standards must include community. Articles such as this are useful for analysing and thinking through some of the structural and conceptual issues that exist, but once the decision has been made to decolonise our institutions different expertise is required. The British Museum's database was constructed primarily by technology staff with little consultation with curators, let alone the source communities from which many items were acquired, and decisions made about what information to include and what to exclude were all internal [79,80]. Involving community requires more than curatorial representation, requests for permission, or periodic consultation sessions. Self-determination requires involvement from the outset [73], at all levels of the process. Otherwise our future museum documentation, like many of our current systems, will continue to privilege white European perspectives, maintaining a legacy of control over artefacts, knowledge, and stories which started when Cook and Banks first landed on Gweagal land nearly 230 years ago.

## 6. Conclusions

Museum documentation is not neutral. The way we describe artefacts and their significance has the potential to promote some perspectives while suppressing or obscuring others. In recent decades ideas about provenance have become more complex, and museums more relational in their thinking; but some museum documentation, particularly of ethnographic and anthropological collections, continues to remain centred on colonial figures and white European perspectives.

Documentation has a role to play in complicating these stories, and in countering the effects of received privilege and perceived authority. Despite the inherent difficulties, and the wealth of legacy data that exists (the provenance for which is itself often poorly documented), collecting institutions need to continue exploring expanded, polyvocal approaches to description. Related disciplines, including archival science, data provenance, and archaeology provide some guidance here, as do the concepts of inclusion, diversity, polyvocality, and participation underpinning the exhibitions and philosophy of many contemporary museums. If combined with community involvement and a commitment to Indigenous self-determination, these elements have the potential to produce more relational documentation, which in turn better reflects the complexities of our collections, and of the world.

**Funding:** This research received no external funding.

**Acknowledgments:** The author would like to thank Nathan Sentance for reading an earlier draft of this paper; and Rebe Taylor for her thoughtful comments leading up to submission.

**Conflicts of Interest:** The author declares no conflict of interest.

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
