# Peer review of "Collections in the Expanded Field: Relationality and the Provenance of Artefacts and Archives"

_heritage, doi:10.3390/heritage2010059_

Round 1
Reviewer 1 Report
This article is timely and well written and I have enjoyed reading it. My comments are below. It has never been explained who gets what comments, so I am submitting all to editors and author.
line 49 “mseuem” should be “museum”
lines 53-59 please give a short example of Krauss’s sculpture subject. I found the abstract idea distracting and unfinished. I understand that it is a foundation of your argument and paper title, but without an example I feel that it teeters on the edge of empty jargon. I do understand the argument for documentation that follows, but I had to back up a couple of times to realize the Krauss reference was unfinished and not telling me something.
Line 315-316 “Of the moment in 1937…” had me looking for a first half of a sentence even though your idea is ultimately complete. May I suggest something like this:
“Thomson also felt an outsider in white Australian society more broadly; he describes the moment when, in 1937, HMAS Moresby came to pick up him and his collection from the Northern Territory: “I felt old, very tired and very dirty.”
line 451-456 You might consider expanding or paring back this section, I am not sure which; it is a little like the Krauss example in my opinion. I agree with what you are saying before this in regard to Indigenous communities and the way they relate to objects, and the need for the flexible metadata you discuss. In this section on “decanted” exhibitions you open up a whole new subject without good resolution in my opinion. There are museums that have instituted, for the public, a messy and “undecanted” “open storage” where shelves of objects in storage are visible to the visitor, with a plexi/glass barrier for safety. For example, Museum of Anthropology at University of British Columbia (UBC) does this with Indigenous collaboration about their own categories AFTER a period of time with unlabeled, uninterpreted open storage. Smithsonian has had some examples at National Museum of the American Indian, specifically the beadwork and projectile point cabinets. The Walters Art Museum has had a display window for watching conservators at work. See links below for other discussions, pros and cons, and pitfalls if you have not encountered this before.
http://www.museumethics.org/2009/09/open-storage/
https://www.artsy.net/article/artsy-editorial-new-york-museums-open-their-storage-to-the-public-putting-their-vast-collections-on-display
Author Response
Thank you to the reviewer for their clear and concise comments, which have helped to tighten and improve the final article.
On the specific points raised:
line 49 “mseuem” should be “museum”
- this has been corrected
lines 53-59 please give a short example of Krauss’s sculpture subject. I found the abstract idea distracting and unfinished. I understand that it is a foundation of your argument and paper title, but without an example I feel that it teeters on the edge of empty jargon. I do understand the argument for documentation that follows, but I had to back up a couple of times to realize the Krauss reference was unfinished and not telling me something.
- I accept that this idea was not fully developed. Rather than distract from the main argument I have removed this section, retaining a small portion in a footnote to acknowledge the origin of the 'expanded field' phrase, which has been retained
Line 315-316 “Of the moment in 1937…” had me looking for a first half of a sentence even though your idea is ultimately complete. May I suggest something like this:
“Thomson also felt an outsider in white Australian society more broadly; he describes the moment when, in 1937, HMAS Moresby came to pick up him and his collection from the Northern Territory: “I felt old, very tired and very dirty.”
- suggestion adopted - this improves the expression of this passage significantly
line 451-456 You might consider expanding or paring back this section
- I agree this is something of a tangent - I have removed the sentence regarding exhibitions from this section, and hope to explore this idea further in my future research. I thank the reviewer for the examples provided.
Reviewer 2 Report
Discussing a case of study the author presents some interesting very well documented opinions about the museums documentation management culturally oriented.
Author Response
Thank you to the reviewer for their time reading the article, and for their comment.